Microgreens: nutritional properties, health benefits, production techniques, and food safety risks

Kaya Seyda s.gungor@alparslan.edu.tr 1
Yardımcı Hülya 2
1 Institute of Health Sciences, Ankara University , Ankara , Turkey
2 Faculty of Health Sciences, Ankara University , Ankara , Turkey
Okpala Charles
Electronic publication date: 2025 Nov 25
Publication date: 2025
Volume: 13
Electronic Location ID: e17938
Received 2023 Aug 25; Accepted 2025 Oct 27
Copyright: ©2025 Kaya and Yardımcı
Copyright year: 2025
Copyright holder: Kaya and Yardımcı
License: This is an open access article distributed under the terms of the Creative Commons Attribution License, which permits unrestricted use, distribution, reproduction and adaptation in any medium and for any purpose provided that it is properly attributed. For attribution, the original author(s), title, publication source (PeerJ) and either DOI or URL of the article must be cited.
License URL: https://creativecommons.org/licenses/by/4.0/

Keywords: Germination, Hydroponic production, Indoor farming, Microgreens, Nutraceuticals

Funding: The authors received no funding for this work.

==============================
“Microgreens” is a hypothetical name given to a new class of edible plants that have become popular in recent years. Microgreens are plants that are larger than sprouts and smaller than baby greens, with an average height of 2–8 cm. They have some advantages, such as the microgreens are better than mature greens in terms of vitamins, minerals, antioxidants and phenolic compounds and the dense and digestible nutrient profile they contain. Microgreens appeal to vegan and vegetarian individuals and simple cultivation in a home environment. Industrial cultivation techniques are being developed under different environmental conditions to ensure that microgreens have a superior nutritional profile. Also, the current health benefits of microgreens are noteworthy. On the other hand, they may pose some food safety risks, especially due to cross-contamination which has created the need to establish suitable decontamination methods for microgreens. Microgreens have become a great topic of interest in the last 10 years, and studies in areas such as nutritional properties and health benefits, production techniques and food safety risks are increasing in the existing literature. This creates the need to bring together existing knowledge about microgreens. Therefore, the aim of this study is to conduct a very detailed literature review about microgreens.

Introduction

A substantial portion of the food produced in the global food chain is subject to deterioration during transportation or storage, and this huge food waste threatens food security globally (Weber, 2017). Moreover, approximately 800 million people worldwide face malnutrition, and the world population is rapidly increasing, so the need for nutritious food production has also increased (Frąszczak & Kleiber, 2022). Existing data show that 60%, 30%, and 15% of the world’s population have micronutrient deficiencies of iron (Fe), zinc (Zn) and selenium (Se), respectively, which is one of the most common nutritional problems in both developed and developing countries (Frąszczak & Kleiber, 2022; Bhaswant et al., 2023). Modern agricultural practices have significant disadvantages in terms of environmental damage, considering the irrigation water and pesticides used. Products such as microgreens should be prioritized to protect the ecosystem and minimize production-related environmental damage (Weber, 2017).

Microgreens are a new class of edible plants that are gaining popularity today (Mir, Shah & Mir, 2017). Lifestyle changes, increasing concerns about health, the orientation toward healthy nutrition and functional foods, as well as the decrease in agricultural lands, and the disadvantages of modern agricultural practices have led to an increased interest in microgreens (Teng, Liao & Wang, 2021; Teng et al., 2023). Microgreens, which have many advantages in terms of nutritional properties, have not been associated with any disease according to the current literature (Teng, Liao & Wang, 2021; Sharma et al., 2022; Teng et al., 2023). With a pleasing color, texture, and flavor, sprouts and microgreens are potential therapeutic and functional foods. They have many positive features such as having a short growth cycle (14 days on average depending on the species) and growing hydroponically as well as in soil or other substrates (Mir, Shah & Mir, 2017; Moraru, Rusu & Mintas, 2022; Sharma et al., 2022; Alloggia et al., 2023). Sprouts and microgreens can be grown in a wide variety of areas, from home balconies and gardens to large farms, can be grown by anyone and do not require professional care (Bhaswant et al., 2023; Teng et al., 2023). It is predicted that microgreens may also contribute to astronauts’ feeding in the future and may benefit the physical and psychological health of the crew during long space flights (Paradiso et al., 2018; Teng et al., 2023). On the other hand, the health effects of radiation and reduced gravity have not yet been fully elucidated. According to data from a noteworthy study, moderate growth retardation and reactive oxygen species (ROS) accumulation were observed in microgreens grown under reduced gravity conditions, likely due to oxidative stress concentrated in the roots. Therefore, further research is needed on this topic (De Francesco et al., 2024).

Microgreens have a good potential to improve and diversify human nutrition with their phytochemical properties (Ebert, 2022). Moreover, unlike baby or mature plants, sprouts and microgreens do not require additional effort and cost such as large tracts of land, soil, fertilization, extra labor, or professional care and can be produced quite simply and quickly (Bhaswant et al., 2023). In addition, microgreens are very suitable food for vegan and vegetarian individuals, and are also very practical for fast city life (Paradiso et al., 2018). Therefore, they have the potential to solve the malnutrition problem (Pathan & Siddiqui, 2022).

As it is known, the basic growth stages of plants are sprouts, microgreens, baby greens and mature greens. Among these, microgreens have stood out in many ways compared to others. Microgreen consumption, which has become increasingly popular all over the world, has become a new focus for the scientific world. Microgreens grow through the germination of seeds in the dark and under high relative humidity. It was reported that the consumption of seeds in the form of sprouts and microgreens dated back to the ancient Egyptians, around 3000 BC (Ebert, 2022). The sprouts are harvested and consumed when the cotyledons are still under-developed and the true leaves are immature, usually in less than a week (3–5 days) when appropriate conditions are provided. The whole sprout-seeds, roots, and shoots-is consumed. On the other hand, microgreens are (average 2–8 cm tall) slightly larger than sprouts, smaller than baby and mature plants, and are harvested within 10–12 days with the development of cotyledons under a light environment (Bhaswant et al., 2023). The stem, cotyledons, and the first true leaves are consumed (Ebert, 2022; Bhaswant et al., 2023).

Microgreens are often used as garnishes in salads, soups, sandwiches, pizzas, pancakes, and appetizers (Mir, Shah & Mir, 2017; Ebert, 2022). They can also be consumed with healthy smoothies, juices, and other beverages (Gupta et al., 2023). The image of the arugula (Eruca vesicaria) microgreen grown by the researchers in the natural day-night cycle in soil is given in Fig. 1.

Figure 1 The image of the arugula (Eruca vesicaria) microgreen grown by the researchers in the natural day-night cycle in soil.

In this review, current literature on the main sources and nutritional properties of microgreens, their health benefits, production techniques, and food safety risks were documented. Microgreens, which is an alternative food for all individuals aiming to eat healthy and those who prefer to eat vegan, vegetarian and similar diets, are also phytochemically advantageous compared to mature greens. In addition, when the necessary production conditions are provided, it can become nutraceutical for people with special needs. Therefore, this study is necessary because it reveals the potential and health advantages of microgreens, which has become one of the popular disciplines of recent years, to be a solution to the search for alternative food and nutritional deficiencies all over the world.

Survey Methodology

Microgreens is a new concept that has been popular in recent years, and the available literature is quite limited. To identify articles related to microgreens, we conducted searches using the keyword ‘microgreens’ in the Science Direct, Scopus, PubMed, Web of Science and Google Scholar databases. After passing abstract screening, the full text of all publications found was downloaded and read in detail. The main subheadings obtained from all studies were as follows, production techniques, health benefits, nutritional properties and food safety risks of microgreens. Therefore, our study presents a detailed literature review of these subheadings. Moreover, studies from the last twenty years were included, and studies that were not related to the basic subheadings mentioned above were excluded and only English and full-text articles were used for data extraction and analysis.

Discussion of Synthesized Literature

Nutritional properties of microgreens

Microgreens have become common with their nutritional advantages and health benefits in in-vivo and in-vitro studies conducted in recent years (Ebert, 2022). They are also very important nutrients sources since they are very rich in antioxidants, vitamins, minerals, glucosinolates, chlorophyll, phylloquinone, and flavonoids and anthocyanins, which are two phenolic compounds (Kyriacou et al., 2022; Alloggia et al., 2023). Microgreens are rich in minerals, such as potassium (K), calcium (Ca), nitrogen (N), phosporus (P), sulfur (S), manganese (mn), selenium (Se), and molybdenum (Mo) (Bhaswant et al., 2023). They also contain plenty of α-tocopherol, beta carotene, ascorbic acid, and phylloquinone (vitamin K1). Brassicaceae microgreens are also reported to be a good source of K, Ca, Fe, and Zn in general (Paradiso et al., 2018). In a recent study, Arya, Sangeeta Kutty & Pradeepkumar (2023) examined the nutritional content of microgreens. In the study, it was reported that the crude protein in microgreen species grown under different conditions (room or rain shelter) varied between 0.0−2.0 g/100 g. The crude fiber rate in microgreens varied between 0–25%, and the highest crude fiber rate was found in wheat microgreens grown in the rain shelter, and the lowest crude fiber rate was found in amaranthus microgreens grown in the room. Additionally, the amounts of vitamin C, beta carotene, Ca and Fe per 100 grams of all microgreen species vary between 20–40 mg, 1.5−3.0 mg, 100–300 mg and 0.2−1.5 mg, respectively. Bhaswant et al. (2024) examined the macro and micronutrients of Perilla frutescens var. crispa f. viridis microgreens and germinated seeds. In the study, an average of total protein was 8.032 ± 0.035 mg/g and total lipid was 5.016 ± 0.044 mg/g in microgreens. In the germinated seed, these values are 97.886 ± 0.366 mg/g and 234.096 ± 0.632 mg/g, respectively. The average moisture content of microgreens is 94.68%, while it is 38.58% in germinated seeds and the mineral contents of microgreens given in Table 1. Til̇ahun et al. (2024) examined the effect of pre-harvest Methyl Jasmonate treatment on the nutritional properties of five different radish microgreens. The mineral contents of the five microgreens control samples that were not treated with Methyl Jasmonate in the study are given in Table 1. On the other hand, it has been reported that the ascorbic acid content of microgreen cultivars varies between 150–200 mg/100 g DW. Interestingly, Castellaneta et al. (2024) investigated the lipid profile of microgreens produced from chia, flax, soy, sunflower and rapeseed, which are notable for their oil content. The study reported that flax and chia microgreens were sources of α-linolenic acid (ALA). It was also reported that the highest amounts of triglycerides and sterol esters were found in rapeseed, sunflower and flax, respectively. Durairajan et al. (2024) investigated the in vitro bio-accessibility of sprouts and microgreens produced from barnyard millet (Echinochloa esculenta (A. Braun) H. Scholz). In the study, some nutritional values of crude sprouts and microgreens were given (Table 1). According to these data, total carbohydrates in sprouts were 367.16 ± 0.54 mg/g, while in microgreens it was 416.81 ± 0.99 mg/g. Total protein values in sprouts and microgreens were found to be 178.55 ± 50 mg/g and 218.29 ± 4.75 mg/g, respectively. In another study, Marchioni et al. (2021) examined some species of the Brassicaceae family (broccoli, radish, mustard, arugula, and watercress) comparatively on the basis of some phytochemical compounds, such as chlorophylls, polyphenols, carotenoids, anthocyanins, ascorbic acid, total reducing sugar, and antioxidant activity. In the study, broccoli was found to have the highest nutrient profile and good antioxidant capacity with the highest polyphenol carotenoid and chlorophyll content, while mustard was characterized by its high ascorbic acid and total sugar content. Arugula was reported to have the lowest antioxidant content.

Table 1 The amounts of nutrients some plant species contain according to their growth stages.

		Digestible carbohydrate	Total carbohydrate	Total soluble Starch	Digestible protein	Total protein	Total lipid	Moisture	Ascorbic acid	Vitamin-A	β-carotene	Ca	K	Zn	Mn	P	Mg	S	Na	Mo	Cu	Fe	Se	Sr	I	
Barnyard millet (Echinochloa esculenta (A. Braun) H. Scholz)
(Durairajan et al., 2024)	Sprouts		367.16 ±0.54 mg/g	220.55 ±0.50 mg/g		178.55 ±50 mg/g			184.43 ±1.70 μg/g	223.53 ±1.72 μg/g																
Microgreen		416.81 ±0.99 mg/g	289.11 ±2.20 mg/g		218.29 ±4.75 mg/g			203.13 ±2.37 μg/g	242.48 ±2.40 μg/g																
(Perilla frutescens var.crispa f. viridis)
(Bhaswant et al., 2024)	Germinated seeds					97.886 ±0.366 mg/ 100 g FW	234.096 ±0.632 mg/g FW	38.58%	ND		ND	247.981 ±11.382 μg/100 g FW	193.339 ±12.002 μg/100 g FW	0.615 ±0.052 μg/100 g FW	2.3477 ±0.126 μg/100 g FW	194.196 ±15.520 μg/100 g FW	157.509 ±4.077 μg/100 g FW	145.629 ±18.476 μg/100 g FW	3.639 ±1.175 μg/100 g FW		0.981 ±0.053 μg/100 g FW	4.061 ±0.163 μg/100 g FW	0.003 ±0.001 μg/100 g FW	1.037 ±0.006 μg/100 g FW	17.804 ±3.240 μg/100 g FW	
Microgreen					8.032 ±0.035 mg/ 100 g FW	5.016 ±0.044 mg/g FW	94.68%	ND		16.146 ±2.017 μg/g FW	47.112 ±0.267 μg/100 g FW	294.839 ±10.839 μg/100 g FW	0.096 ±0.000 μg/100 g FW	0.376 ±0.009 μg/100 g FW	30.018 ±2.056 μg/100 g FW	26.092 ±1.093 μg/100 g FW	29.870 ±9.973 μg/100 g FW	1.216 ±0.055 μg/100 g FW		0.118 ±0.003 μg/100 g FW	0.617 ±0.060 μg/100 g FW	0.0003 ±0.000 μg/100 g FW	0.096 ±0.000 μg/100 g FW	1.727 ±0.162 μg/100 g FW	
Radish microgreens
(Til̇ahun et al., 2024)	Cultivar 1											4,040 ±12 mg/kg DW	7,890 ±4 mg/kg DW	31.12 ±0.1 mg/kg DW	23.6 ±0.1 l mg/kg DW	9,459 ±8 mg/kg DW	3,474 ±21 mg/kg DW		1,414 ±3.2 mg/kg DW	1.51 ±0.1 mg/kg DW						
Cultivar 2											3,547 ±63 mg/kg DW	9,568 ±55 mg/kg DW	27.89 ±1.1 mg/kg DW	30.9 ±2.4 mg/kg DW	10.575 ±67 mg/kg DW	3,548 ±23 mg/kg DW		1,208 ±7.1 mg/kg DW	0.26 ±0.4 mg/kg DW						
Cultivar 3											3,536 ±64 mg/kg DW	10.372 ±59 mg/kg DW	27.32 ±1.1 mg/kg DW	18.6 ±2.5 mg/kg DW	10.251 ±72 mg/kg DW	3,511 ±25 mg/kg DW		1,121 ±6.8 mg/kg DW	0.78 ±0.4 mg/kg DW						
Cultivar 4											4,698 ±3 mg/kg DW	7,221 ±4 l mg/kg DW	27.65 ±0.1 mg/kg DW	25.6 ±0.1 mg/kg DW	10.935 ±5 mg/kg DW	3,404 ±20 mg/kg DW		1179 ±3.7 mg/kg DW	0.08 ±0.1 mg/kg DW						
Cultivar 5											5,034 ±42 mg/kg DW	8,526 ±65 mg/kg DW	28.55 ±0.23 mg/kg DW	25.3 ±4.1 mg/kg DW	10.806 ±13 mg/kg DW	3,433 ±14 mg/kg DW		1,194 ±3.1 mg/kg DW	0.12 ±0.2 mg/kg DW						
Tropical spinach
(Amaranthus sp.)
(Ayeni, 2021)	Micro/Baby Green
(100 g DM)	%17			%33.5		%2					%0.9	%6.7	170 ppm	>260 ppm											
Field Grown Mature Foliage (100 g DM)	%27			%25		%0.5					% 2.9	%4.1	<40 ppm	<50 ppm											
Roselle plant
(Hibiscus sabdariffa)
(Ayeni, 2021)	Micro/Baby Green
(100 g DM)	%24			%30		%2					%0.9														
Field Grown Mature Foliage (100 g DM)	%37			%22.5		%1					%1.7														
Red Salanova butterhead lettuce
(Lactuca sativa L. var. Capitata)
(El-Nakhel et al., 2020)	Microgreen											59.71 ±0.88 mg 100 g−1 FW	127.4 ±5.03 mg 100 g−1 FW			12.25 ±0.44 mg 100 g−1 FW	22.95 ±0.60 mg 100 g−1 FW	6.85 ±0.20 mg 100 g−1 FW	32.56 ±0.91 mg 100 g−1 FW							
Mature Leaves											21.85 ±0.73 mg 100 g−1 FW	430.4 ±3.76 mg 100 g−1 FW			19.28 ±0.89 mg 100 g−1 FW	10.50 ±0.21 mg 100 g−1 FW	4.56 ±0.26 mg 100 g−1 FW	4.02 ±0.12 mg 100 g−1 FW							
Green Salanova butterhead lettuce
(Lactuca sativa L. var. Capitata)
(El-Nakhel et al., 2020)	Microgreen											52.48 ±0.85 mg 100 g−1 FW	117.3 ±2.33 mg 100 g−1 FW			12.33 ±0.58 mg 100 g−1 FW	23.62 ±0.61 mg 100 g−1 FW	6.51 ±0.20 mg 100 g−1 FW	33.42 ±1.39 mg 100 g−1 FW							
Mature leaves											29.17 ±0.34 mg 100 g−1 FW	419.8 ±8.18 mg 100 g−1 FW			18.94 ±0.93 mg 100 g−1 FW	12.49 ±0.37 mg 100 g−1 FW	3.03 ±0.03 mg 100 g−1 FW	4.86 ±0.18 mg 100 g−1 FW							
Jute(Corchorus olitorius)
(Ebert, 2022)	Microgreens								34.9 mg/100 g FW																	
Mature leaves								10.0 mg/100 g FW																	
Cucumber(Cucumis sativus)
(Ebert, 2022)	Microgreens								25.0 mg/100 g FW																	
Mature leaves								17.45 mg/100 g FW																	
Red cabbage
(Ebert, 2022)	Microgreens										11.5 mg/100 g FW															
Mature leaves										0.044 mg/100 g FW															
Notes.

DM dry matter

FW fresh weight

ND not detected

Sprouts and microgreens are more digestible than seeds (Kyriacou et al., 2022). During the germination process of the seed, hydrolytic enzymes are activated. They increase the bioavailability of the product by separating the nutrients from the phytate chelates. The bioavailability of particularly Fe and Ca and the digestibility of proteins increase as a result of germination and microgreen growth and the inactivation of phytate, oxalate, and tannins (Ebert, 2022; Kyriacou et al., 2022). Ascorbic acid, tocopherol, beta carotene, and phylloquinone levels increase in microgreens via photosynthesis (Ebert, 2022).

Many studies have proven that microgreens have a better nutrient profile than mature plants. For example, Ayeni (2021) compared the nutrient content of microgreens/baby leaves of tropical spinach (Amaranthus sp.) and roselle plant (Hibiscus sabdariffa L.) with mature leaves. Compared to mature leaves grown in the field, greenhouse-grown micro/baby greens were found to be richer in digestible proteins, P, K, Mg, Fe, Mn, and Zn, although poorer in digestible carbohydrates and Ca. According to the results of the study, microgreens (32.5%) in tropical spinach contain 7.5% higher digestible protein than mature leaves (25%). At the same time, spinach microgreens contain 6.7% potassium (K), while this rate is 4.1% in mature leaves. P, Fe and Mg values in tropical spinach were not affected by growth stages. Mn and Zn are >260 ppm and 170 ppm, respectively in spinach microgreens, while they are <50 ppm and <40 ppm, respectively in mature leaves. On the other hand, in the Roselle plant, the significant difference was more evident, and it was reported that all macronutrients except Ca were found in higher amounts in microgreens compared to mature leaves. Weber (2017) reported that broccoli microgreens contained more Mg, Mn, Cu, and Zn than mature plants. Microgreens are known to have four to 40 times higher concentrations of carotenoids than mature plants (Frąszczak & Kleiber, 2022). In a study by Paradiso et al. (2018), comparing the estimated daily intake values of nutritional minerals obtained from the consumption of microgreens and mature lettuce, considering the average lettuce consumption of 22.5 grams per person (in Europe), microgreens were superior. Ca, Mg, Fe, Mn, Zn, Se and Mo contents of lettuce microgreens were reported as 34.8 mg/day, 6.5 mg/day, 0.32 mg/day, 0.28 mg/day, 0.13 mg/day, 1.49 µg/day, 2.58 µg/day, respectively. In mature lettuce, these values were reported as 17.2 mg/day, 4.5 mg/day, 0.17 mg/day, 0.03 mg/day, 0.08 mg/day, 0.28 µg/day, 1.04 µg/day, respectively. Zou et al. (2021) comparatively investigated nutritional metabolites in Chinese cabbage (Brassica rapa subsp. chinensis var. Parachinensis) during the growth process from shoot to mature leaves. In the study, while essential amino acids, folate, and β-cryptoxanthin decreased from sprouts to mature leaves, the total amount of reducing sugar, carotenoids, and vitamin K1 increased. It was also reported that most of the important minerals were concentrated in microgreens. Microgreens have a higher nutritional value compared to mature plants, which is associated with the fact that these nutrients are more intense nutrient sources (Weber, 2017). Compared to the nutrient concentration in mature leaves, the leaves of microgreens were found to have a higher nutrient content (Xiao et al., 2012). The amounts of nutrients some plant species contain according to their growth stages are given in Table 1. When datas from limited studies are evaluated independently of the cultivar, microgreens contain, on average, 15–25% digestible carbohydrates, 40–45% total carbohydrates, 25–30% total soluble starch, 30–35% digestible protein, 8–25% total protein and 0.5–2% total lipid in terms of macronutrients. In addition, there is an average of 90–95% moisture of microgreens. These data may deviate moderately when the dry and fresh weight parameters of the material are considered. On the other hand, the protein and lipid profile of the seed has a direct effect on the nutritional values of microgreens. Therefore, genotype is an important factor in terms of the nutritional values of microgreens. Micronutrients in microgreens are quite variable and it is difficult to make an average inference. Further studies are necessary to provide a broader perspective on the average nutritional values of microgreens under all environmental and biological conditions.

The raw consumption of microgreens and not exposing them to heat treatment prevents losses in heat-sensitive vitamins such as ascorbic acid. Similarly, the ratio of chlorophyll and phenolic compounds with antioxidant activity is higher in microgreens compared to sprouts (Ebert, 2022). Microgreens are also characterized by high absorption of these bioactive compounds (Frąszczak & Kleiber, 2022). El-Nakhel et al. (2020) comparatively investigated the phytochemical and antioxidant properties of green and red-pigmented lettuce (Lactuca sativa L. var. capitata cultivars) at different developmental stages. In the study, Ca and Mg were found to be more intense in microgreens than mature leaves, regardless of cultivar. Total polyphenols were higher in microgreens, especially in the red-pigmented lettuce cultivar. In addition, it was reported that this density increased in red-pigmented lettuce with higher levels of P, K, nitrate, chlorophyll, lutein, and betacarotene in mature leaves.

Biological enrichment processes applied to foods can also be applied to microgreens. Biofortification is a sustainable approach that increases the bioavailability of nutrients for humans by improving the nutritional quality of plants and has the potential to offer long-term solutions to minimize food insecurity and malnutrition (Ma et al., 2022). Frąszczak & Kleiber (2022) studied the effect of the iron chelate-fortified solution on plant growth and mineral concentrations of some microgreens (purple kohlrabi, radish, peas, and spinach). In the study, it was reported that Fe content increased in the leaves of all species except radish, but this increase reduced the ratio of zinc and copper regardless of genotype. Puccinelli et al. (2019) reported that when microgreens produced from basil seeds enriched with selenium were included in human nutrition, they could support general immunity by contributing to the antioxidant defense system. Newman et al. (2021) reported that the total phenol content and antioxidant capacity could be increased by enriching basil, coriander, and green onion microgreens with selenium. Kathi et al. (2023) fed broccoli microgreens grown hydroponically on polyethylene pads with different concentrations of ascorbic acid solutions (0% (control), 0.05%, 0.1%, 0.25%, and 0.5%) and examined the effect of the treatment on nutrient composition. In the study, it was reported that ascorbic acid solution increased biomass, carotenoids, chlorophyll amount, potassium levels, especially ascorbic acid in microgreens, but that this increase in ascorbic acid showed a negative correlation with minerals N, P, Mg, Ca, and S. Therefore, further studies are needed for the optimization of genotype-specific nutrient enrichment processes.

In addition to their highly valuable nutritional compositions, microgreens also meet consumer demands in terms of their good organoleptic properties, such as sensory intense taste, lively texture, pleasant aroma, and beautiful appearance. Depending on the variety, their taste can be bitter, sour, mild, or spicy (Bhaswant et al., 2023). Michell et al. (2020) presented six types of microgreens (arugula, broccoli, bull beet, red cabbage, red garnet amaranth, and pea sprouts) to the sensory perception of the consumer and examined their acceptability in a consumer panel of 99 individuals. In the study, it was reported that all species received high average ratings of appreciation from the participants and broccoli, red cabbage and peas were liked the most. The mean ranged from highly acceptable to slightly acceptable.

Health benefits of microgreens

Antimicrobial, anti-inflammatory, antioxidant, and anticarcinogenic effects of microgreens have been reported (Gupta et al., 2023). Microgreens containing plenty of anthocyanins can alleviate chronic low-grade inflammation, especially due to obesity (Lee et al., 2017). Tallei et al. (2025) investigated the anti-inflammatory, anticancer, and antioxidant properties of red radish microgreen extract using both in silico and in vivo approaches (with Drosophila melanogaster larvae). In silico findings showed that the extract interacted with key proteins such as MAPK1 and AKT1, which regulate pathways related to cancer, inflammation, and oxidative stress. In vivo results indicated that the extract suppressed immune-related genes and reduced ROS levels through the NFκB and JAK-STAT pathways.

In addition, studies on rats have shown that they are highly effective in improving diabetes mellitus (DM) symptoms and increasing insulin sensitivity, reducing body weight, and modulating intestinal microbiota and lipid metabolism (Huang et al., 2016; Li et al., 2021; Mohamed et al., 2022; Khattab et al., 2022; Ma et al., 2022). Sosnowska et al. (2025) examined the bioactive properties and phenolic profile of red radish microgreens after in vitro digestion. They also evaluated the post-digestion effects of these microgreens on Caco-2, HT-29, RAW 264.7, and SH-SY5Y cell lines. They found a higher bioavailable fraction after gastric digestion compared with small intestinal digestion, although microgreens that were not digested in the small intestine did not exert cytotoxic effects on the cell lines. The results revealed that the microgreens inhibited ROS formation, demonstrated antioxidant activity, and exerted a protective effect by suppressing nitric oxide production.

Mohamed et al. (2022) investigated the therapeutic effects of barley microgreens added to the diet in streptozotocin-induced diabetic rats with and without aflatoxin. In the study, streptozotocin injection and/or aflatoxin administration significantly increased glucose levels, decreased insulin secretion and beta cell functions, increased oxidative stress, and caused deterioration in liver and kidney functions. The addition of barley microgreens to the diet, independent of streptozotocin, improved all these parameters. Similarly, Nakakaawa et al. (2025) investigated the therapeutic effects of Brassica carinata microgreen extract in diabetic rats. They reported that the extract did not cause hypoglycemia in rats with normal blood glucose levels, whereas it exhibited antidiabetic effects in rats with experimentally induced type 2 diabetes. These effects included lowering fasting blood glucose, improving oral glucose tolerance, increasing insulin sensitivity, and reducing insulin resistance. In another study, Li et al. (2021) reported that broccoli microgreen juice given to high-fat diet-induced obese C57BL/6J mice for 10 weeks promoted body weight loss by increasing insulin sensitivity and modulating intestinal microbiota. In the study, it was reported that broccoli microgreen juice significantly increased the relative abundance of Bacteroidetes in the microbiota composition and decreased the ratio of Firmicutes to Bacteroidetes, which, in turn, contributed to the production of short-chain fatty acids (SCFA), which are known to have anti-inflammatory effects, in the intestines. Garcia-Perez et al. (2025) examined the effects of Brassicaceae microgreens on gut bacteria. They observed a significant correlation between Parabacteroides merdae, a gut bacterium with beneficial effects on human health, and isothiocyanates. Furthermore, these microgreens were associated with increased short-chain fatty acid and pseudocytokine production in the colon. Ma et al. (2022) investigated the effect of lyophilized broccoli microgreens on microbiota, blood lipids, inflammatory factors, and hypoglycemia in rats. In the eighteen-week study, body weight, glucose homeostasis, blood lipid parameters, antioxidant indices, and inflammatory biomarkers improved and intestinal microbiota composition developed positively following the consumption of broccoli microgreens. The authors noted that microgreens might specifically improve symptoms of type 2 DM. Similarly, in another eight-week study, it was reported that red cabbage microgreens increased body weight, and decreased LDL levels, triglyceride, and cholesterol levels in rats (n:60, male) given a high-fat diet (Huang et al., 2016). These results suggest that microgreens can reduce body weight gain, regulate cholesterol metabolism, and thus be protective against cardiovascular disease risks. In addition, Khattab et al. (2022) reported that powdered and pelleted barley (Hordeum vulgare L.) microgreens reduced reproductive dysfunction and oxidative stress in streptozotocin-induced diabetes and aflatoxicosis in male mice.

Microgreens are also considered a nutraceutical factor. Renna et al. (2018) reported that low K-containing microgreens could be produced for patients with renal dysfunction. In the study, chicory (Molfetta) and lettuce (Bionda da taglio) microgreens were grown hydroponically on polyethylene pads with phytonutrient solutions containing K at various concentrations. Regardless of genotype, microgreens grown with nutrient solutions containing less K successfully contained less K and more nutrient-dense value. The authors reported that the production of low-K microgreens could be a therapeutic strategy in individuals with kidney damage. In another study, Jambor et al. (2025) investigated the effects of Trigonella foenum-graecum L. microgreens on TM3 murine Leydig cells. They reported that ROS formation was significantly suppressed in these cells, particularly after 24 h of exposure, and that dose-dependent positive effects were observed on cell membrane integrity and mitochondrial membrane potential. Furthermore, progesterone and testosterone production were stimulated at doses up to 250 µg/mL, whereas a decrease in steroid release was observed with higher doses. Based on the results of this in vivo study, it can be concluded that microgreens may contribute to reproductive health due to their polyphenolic compounds.

On the other hand, although quite limited, some studies have been conducted to investigate the effectiveness of microgreens in cancerous cells. Fuente et al. (2020) evaluated the antiproliferative effect of four hydroponically grown Brassicaceae family microgreens (broccoli, cabbage, mustard, and radish) between normal and colon cancer cells in vitro. While microgreens did not show any effect in normal cells, they showed an antiproliferative effect in cancerous cells. Therefore, daily consumption of microgreens with a balanced diet may be a preventive nutritional strategy to reduce the burden of chronic degenerative diseases such as colon cancer.

No side effects of microgreens have been reported in studies to date. In a study on this subject, the oral acute (14 days) and subacute (28 days) toxicity of ethanol extract of Brassica carinata A. Braun microgreens, which are a type of Ethiopian mustard, was investigated in Wistar rats. In both acute and subacute toxicity studies, no death, signs of abnormality, or intervention-related adverse events were observed at doses of 2,000 mg/kg, 1,000 mg/kg, 500 mg/kg, and 250 mg/kg. The liver, kidney, lungs, and heart were normal when they were examined histopathologically. In the study, it was reported that the extract was safe and non-toxic and that it was used in various regions of Sub-Saharan Africa for gastrointestinal disorders (Nakakaawa et al., 2023).

Production techniques of microgreens

Microgreens can be produced from the seeds of a wide variety of plants, such as legumes (i.e., chickpeas, lentils, soybeans, mung beans, and black-eyed peas), grains (i.e., barley, rye, corn, oats, and rice), pseudo cereals (i.e., quinoa and buckwheat), oilseeds (i.e., sunflower seeds, hazelnuts, flax seeds, sesame, and almonds), and vegetables (i.e., beetroot, radish, arugula, cress, fenugreek, basil, spinach, onion, leek, celery, lettuce, mustard, cabbage, and broccoli) (Ebert, 2022). In addition, although it is not used frequently, microgreens can be produced from many wild species. Due to their phytochemical richness, species belonging to the Brassicaceae family are more preferred and come to the fore (Renna et al., 2018).

Today, microgreens can be grown in simple ways at home and in various environmental conditions in the food industry (Bhaswant et al., 2023). Fluorescent tubes or high-pressure sodium lamps are frequently used artificial lighting systems, especially in indoor farming systems. However, in recent years, light-emitting diodes (Light Emitting Diode-LED) have been used more widely due to their advantages such as energy saving, minimum heat transfer to the seed, and durability (Alloggia et al., 2023). In addition, with the development of LED technology, optimization of spectral quality (wavelength), intensity (photon flux), and photoperiod, which can increase the phytochemical content of sprouts and microgreens, have become possible (Kyriacou et al., 2022). Lighting with various wavelengths from the light spectrum (from ultraviolet (UV) to infrared) can affect the biosynthesis of phytochemicals in different ways (Artés-Hernández, Castillejo & Martínez-zamora, 2022). Wavelength is very important to stimulating secondary metabolites by modulating the expression of specific genes (Appolloni et al., 2022). Plants are exposed to abiotic (UV, light, flood, drought, salinity, heavy metals, extreme heat, injuries, etc.) and biotic (bacteria, fungi, insects, and other small animals) stress factors during their growth stages. Under the influence of these stress factors, the transcriptional factors that form the plant’s defense mechanism against stress are triggered (Artés-Hernández, Castillejo & Martínez-zamora, 2022).

Polyphenols called ’secondary metabolites’ produced by plants against stress factors have subgroups, such as phenolic acids, flavonoids, and anthocyanins. These compounds exhibit antioxidant and anti-inflammatory properties as well as free radical scavenging or antimicrobial activities (Appolloni et al., 2022). In an experimental study, radish microgreens were reported to be among the highest-yielding crops in commercial production. It was found that as the β-carotene content in the microgreens increased, their appearance improved, and as the magnesium content increased, their taste and texture also improved (Rebolledo et al., 2024).

Changes in phytochemical compounds can be observed according to the type of UV to which plants are exposed (Appolloni et al., 2022). In a study, Bucky et al. (2024) examined the effects of varying levels of red and blue LED light on the growth and antioxidant properties of red amaranth (Amaranthus tricolor L.) and red lettuce (Lactuca sativa L.) microgreens. The results showed that higher proportions of red light promoted biometric responses such as longer stems and leaves, whereas greater amounts of blue light increased the levels of total phenols, antioxidant compounds, betalains, and anthocyanins. Brazaitytė et al. (2019) studied the optimum growth and phytochemical content of mustard microgreens (Brassica juncea L. cv.) by growing them under UV-A light emitting LEDs of different wavelengths (366 nm, 390 nm, 402 nm) and photoperiods (10 and 16 h). In the study, the most positive effect on mineral deposition in general, except for Fe, was observed at longer UV-A wavelengths (390 nm, 402 nm) in the 16-hour photoperiod. It was noted that lutein/zeaxanthin and β-carotene content increased in response to the shortest UV-A wavelength (366 nm) in the 10-hour photoperiod and a longer UV-A wavelength (390 nm) in the 16-hour photoperiod. On the other hand, Lu et al. (2021) studied the effect of preharvest UV-B treatment on glucosinolate levels in broccoli microgreens (Brassica oleracea var. Italica) grown on a hydroponic pad. In the study, it was reported that the shelf life of broccoli microgreens that had been applied pre-harvest UV-B and calcium chloride spray could extend up to 21 days, there was no significant change in glucosinolate levels in this process, and that the general nutritional quality was preserved. On the other hand, when preharvest UV-B and postharvest UV-B interventions were compared, it was stated that preharvest intervention significantly increased glucosinolate biosynthesis genes and decreased the expression of myrosinase, a gene responsible for the degradation of glucosinolate.

Skowron et al. (2025) examined the effects of UV-B (311 nm) and UV-C (254 nm) exposure applied during the last four days before harvest on green (cv. Sweet Large) and purple (cv. Dark Opal) basil microgreens grown indoors. They found that UV-C significantly impaired photosynthesis and growth, restricting biomass accumulation. In contrast, UV-B exposure significantly increased the antioxidant capacity, total flavonoid, anthocyanin, and ascorbic acid levels in green-leaf basil, although it decreased the phenolic composition in purple-leaf basil. Overall, the study emphasized the generally positive, though species-dependent, effects of UV-B and the negative morphological effects of UV-C on microgreens. In another study on basil microgreens (Ablagh genotype and cv. Violeto, Kapoor, and Red Rubin), Fayezizadeh et al. (2024) investigated the effects of LED light of various spectra on secondary metabolites and biomass. The highest total chlorophyll, chlorophyll a, and nitrate contents were observed in the Violeta cultivar under blue light. Red light increased starch content in the Red Rubin cultivar and flavonoid content in the Violeta cultivar, while the highest biomass was recorded in the Ablagh genotype. Antioxidant capacity and phenolic composition varied among cultivars under different light conditions. Overall, the study demonstrated that the Ablagh genotype was the most productive in terms of antioxidant capacity, vitamin C, and phenolic compounds under continuous blue light. Similarly, Thongtip et al. (2024) reported that green light exposure promoted taller plant growth in all basil species (peppermint, Thai basil, cumin, lemon basil, and green holy basil), whereas blue light increased plant width in some species. While white and red light were generally associated with higher total phenolic content, blue light resulted in the highest phenolic content at specific time points, with some fluctuations observed. In another study, Silva et al. (2025) investigated the effects of white, red, and blue LED light, as well as UV-C, on the gastrointestinal digestibility of glucosinolates and isothiocyanates in two radish microgreen cultivars (Raphanus sativus; cv. Daikon and cv. Red Rambo). They concluded that the LED light type was a more influential factor than UV-C, and that the combined use of red and blue LED light generally increased glucosinolate content in the Daikon cultivar and isothiocyanate content in the Red Rambo cultivar. Furthermore, significantly higher sulforaphane levels were observed in the Red Rambo cultivar compared with other cultivars after gastrointestinal digestion.

In an experimental study on the effects of light intensity and far-red LED light on broccoli microgreens, Shahkoomahally et al. (2025) reported that hypocotyl length increased with decreasing light intensity. At higher light intensities (100–150 µmol m−2 s−1 PPFD), chlorophyll, carotenoid, and anthocyanin contents increased, whereas at lower intensities (50–75 µmol m−2 s−1 PPFD), ascorbic acid and total phenolic contents were higher. Conversely, the addition of far-red light increased plant height, weight, and antioxidant capacity, but caused a decrease in total phenolic content. Similarly, Silva et al. (2024) examined the effects of different light spectra on lentil (Lens culinaris) microgreens. They found that red, white, and RBW (red–blue–white) lights were advantageous in terms of biomass and energy yield, and that red light, in particular, promoted the accumulation of bioactive compounds such as carotenoids.

Light conditions are of strategic interest in terms of the phytochemical composition of microgreens (Kyriacou et al., 2022). Many studies have been conducted to examine the effects of infrared, red, blue, and ultraviolet rays on the ontogeny and chemical composition of plants. It has been reported that blue light, especially at 400–500 nm, makes photosynthesis efficient while making the plant shorter, thicker and darker green (Artés-Hernández, Castillejo & Martínez-zamora, 2022). In addition, it has been reported that in some genotypes, blue light can increase the production of phenolic acids, flavonoids, carotenoids, anthocyanins, anthocyanidins and some vitamins (E and C) (Appolloni et al., 2022; Artés-Hernández, Castillejo & Martínez-zamora, 2022). Liu et al. (2022) comparatively investigated the growth, nutritional quality, and antioxidant properties of two microgreens of the Brassicaceae family (Brassica oleracea L./cabbage and Brassica alboglabra bailey/Chinese kale) under different light conditions. The light conditions included red:blue:green light emitting LEDs (1:1:1), 30, 50, 70, and 90 µmol m−2 s−1 PPFD values (Photosynthetic Photon Flux Density/PPFD), and 12, 14, 16, 18, 20-hour photoperiods. In the study, it was reported that the hypocotyl length was shortened in both species with the increase in light intensity and that the chlorophyll, carotenoid, soluble sugar, soluble protein, ascorbic acid, and antioxidant capacity of cabbage microgreens increased in the 14-hour photoperiod at 90 µmol m−2 s−1 PPFD. These optimum data were observed in Chinese kale microgreens at a PPFD of 70 µmol m−2 s−1 in a 16-hour photoperiod. Yet, little is known about the effects of different LED light wavelengths and intensities or photoperiods on the growth and nutritional quality of Brassicaceae microgreens. In another similar study, Gao et al. (2021) examined the growth and nutritional quality of broccoli microgreens under red:blue:green light emitting LEDs (1:1:1) at 30, 50, 70, and 90 µmol m−2 s−1 PPFD. In the study, it was reported that the most suitable light intensity for the production of broccoli microgreens was 50 µmol m−2 s−1 PPFD but that the best level of phytochemical accumulation occurred at 70 µmol m−2 s−1 PPFD. On the other hand, glucosinolates decreased significantly as the light intensity increased from 30 µmol m−2 s−1 to 50 µmol m−2 s−1, while it increased significantly as the intensity of the light increased from 50 µmol m−2 s−1 to 90 µmol m−2 s−1. Therefore, to produce microgreens of the best nutritional quality, more studies are needed for the optimization of light conditions, which have quite complex effects, according to different plants. In addition, there are studies conducted to test different wavelengths. Samuolienė et al. (2019) grew microgreens of the Brassicaceae family (kohlrabi, broccoli, and mizuna) in different light spectral qualities (yellow-orange), in a 16-hour photoperiod, and at 300 µmol m−2 s−1 PPFD and examined the levels of insoluble sugar (hexose and sucrose), ascorbic acid, beta carotene, non-heme iron, magnesium, and calcium. In the study, the soluble carbohydrate content of mizuna and broccoli microgreens increased significantly under yellow light at 595 nm, while Fe, Mg, and Ca increased significantly in all microgreens at 622 nm in orange light. The authors reported that the accumulation of Fe in microgreens was highly dependent on promoters and inhibitors of Fe and that they associated positive correlations of Fe-Ca and Fe-Mg in kohlrabi and broccoli microgreens and negative correlation of Fe-beta carotene and Fe-soluble carbohydrates in kohlrabi microgreens with this.

In addition to light, factors such as substrate, temperature, humidity, and seed genotype can also affect the nutritional quality, sensory properties, storage, and shelf life of microgreens (Kyriacou et al., 2022; Gupta et al., 2023). Plants need a substrate that will provide them with attachment, water, oxygen, and nutrients while sprouting. Although this substrate is the soil in traditional agriculture, industrial by-products such as perlite, vermiculture, polyethylene foam, compost, rock wool, coconut fiber, sugar cane fiber, and other various substrate alternatives have been developed with the advancement of technology. In addition to all these, biopolymer-growing fibers are widely used in the production of microgreens (Gupta et al., 2023).

Studies into the effect of substrates on the nutritional value of microgreens are very limited (Alloggia et al., 2023). Saleh et al. (2022) studied the growth and biochemical compositions of kale (Brassica oleracea L. var. acephala), chard (Beta vulgaris var. cicla), arugula (Eruca vesicaria ssp. sativa), and Chinese kale (Brassica rapa var. chinensis) grown under different substrate conditions. In the study, it was reported that the most productive substrate, in general, was the one containing mostly mushroom compost (30% vermicast + 20% sawdust + 20% perlite + 30% mushroom compost). In another study, Kyriacou et al. (2020) studied the effect of natural fiber substrates (agave fiber, coconut fiber, and peat moss) on the nutritional values and phytochemical composition of coriander, kohlrabi, and Chinese kale microgreens compared to synthetic substrates (capillary mat and cellulose sponge). In the study, it was reported that peat moss improved phytochemical composition without sacrificing product performance compared to other substrates and that natural fiber substrates increased macro-micronutrient concentrations in microgreens compared to synthetic substrates. In another study, broccoli microgreens were grown using different growing methods, and their nutritional values were examined. In the study, it was stated that microgreens grown in vermiculture compost contained more minerals than microgreens grown hydroponically (Weber, 2017).

Some studies have shown that geographic location can also change nutritional quality parameters in microgreens. Priti et al. (2021) compared the microgreens of mung beans and lentils grown at low and high altitudes in terms of ascorbic acid, tocopherol, carotenoids, flavonoid, total phenolics, peroxide activity, proteins, enzymes (peroxidase and catalase), and micro and macronutrient contents. Most of the parameters studied at high altitudes in the study, which were observed to have higher temperature amplitude, more photosynthetic active radiation, and UV-B, were superior.

Optimum nutrient content can be achieved in microgreens with appropriate genotype-specific light intensity, wavelength, and photoperiod (Liu et al., 2022). Today, many studies have focused on optimizing species-specific light parameters for the best nutritional quality in microgreens (Alloggia et al., 2023). According to the available literature, the responses and mechanisms of microgreen genotypes to light conditions are not fully known. However, coincidental similarities in the reported results allow us to make some inferences (Artés-Hernández, Castillejo & Martínez-zamora, 2022). Studies on production techniques of microgreens under different conditions are given in Table 2. More experimental studies are needed to understand the optimum production conditions that should be applied for the best nutritional quality in microgreens.

Table 2 Studies on production techniques of microgreens under different conditions.

	Lighting technique and location	Wavelength	Photoperiod	Light intensity	Substrate	Altitude	Temperature	Humidity	
Mustard(Brassica juncea L. cv.)
(Brazaitytė et al., 2019)	LED
Plant growth chamber	Blue: 447 nm
Red: 638 nm
Deep-red: 665 nm
Far-red: 731 nm	10, 16 h	298–300 μ mol m−2s−1PPFD	Hydroponics		21-17 ±2 °C	%50–60 relative humidity	
Broccoli(Brassica oleracea var. İtalica)
(Lu et al., 2021)	Plant growth chamber	UV-B: 312 nm	12 h and additionaly 2 h UV-B per day	42 m−2s−1PPFD	Hydroponics				
Brassicaceae(Brassica oleracea L and Brassica alboglabra bailey)
(Liu et al., 2022)	LED blue/red/green (1:1:1)
Plant growth chamber	Ranged between 400-700 nm	12, 14, 16, 18, 20 h	30, 50, 70 ve 90 μ mol m−2s−1PPFD	Hydroponics				
Broccoli (Brassica oleracea var. italica cv. Lvhua)
(Gao et al., 2021)	LED blue/red/green (1:1:1)
Plant growth chamber	Ranged between 400-700 nm	12 h	30, 50, 70 and 90 μ mol m−2s−1PPFD	Hydroponics		22 ±2 °C	%60 ±5 relative humidity	
Kohlrabi (Brassica oleracea), Broccoli (Brassica oleracea), Mizuna (Brassica rapa)
(Samuolienė et al., 2019)	LED (blue-red, blue-red-green, blue-red-yellow, blue-red-orange)	Blue:447 nm
Red: 638 nm
Red2:635 nm
Far-red:731 nm
Green:520 nm
Yellow:595 nm
Orange:622 nm	16 h	300 μ mol m−2s−1PPFD	Peat substrate		21/17 ±2 °C	%50–60 relative humidity	
Kale (Brassica oleracea L. var. acephala), Chard(Beta vulgaris var. cicla), Arugula (Eruca vesicaria ssp. Sativa) and Chinese kale(Brassica rapa var. chinensis)
(Saleh et al., 2022)	High-pressure sodium lamp
Greenhouse		16 h	300 μ mol m−2s−1PPFD	Mixed in different proportions PittMoss, vermicast, sawdust, mushroom compost, perlite, Pro-mix BX™		24-22 °C	71% mean relative humidity	
Coriander (Coriandrum sativum L.), Kohlrabi (Brassica oleracea L. var. Gongylodes), and Chinese kale (Brassica rapa L.)
(Kyriacou et al., 2020)	LED		12 h	300 ±10 μ mol m−2 s−1 PPFD	Natural (agave fiber, coconut fiber, peat moss) and synthetic (capillary mat, cellulose sponge)		22/18 ±1 °C	%65–%75 ±%5 relative humidity	
Broccoli (Brassica oleracea var.)
(Weber, 2017)	Ecolux bulbs for plant and aquarium		Constant light	Ranged from 3,790 to 4,920 lux	Vermicompost vs Hydroponic growing pads 				
Mung beans and Lentils
(Priti et al., 2021)	Greenhouse		Natural day-night cycle		Coco peat: Vermiculite: Sand (2:1:1) for both	228 m vs 3,500 m	28/26 °C for mung beans and 21/18 °C for lentils		
Radish (Raphanus sativus, cv. Daikon and cv. Red Rambao)
(Silva et al., 2025)	LED	Blue:455 nm
Red: 660 nm
White: 400–700 nm	16 h	400 μ mol m−2 s−1 PPFD	Organic germination substrate (SIRO, Portugal)		22 °C during the day, 19 °C at night	%60 relative humidity	
Amarant (Amaranthus tricolor L.) and Lettuce (Lactuca sativa L.)
(Bucky et al., 2024)	LED	Blue:451 nm
Green: 521 nm
Red: 660 nm
Far-red: 730 nm	18 h	255 μ mol m−2 s−1 PPFD	Coconut coir: vermiculite (3:1)		25 °C	%70–75 relative humidity	
Kale (Brassica oleracea L. cv. Red Russian) and Mustard (Brassica juncea. cv. red lace)
(Gudziṅskaite et al., 2024)	LED
Deep-Red: Blue: White: Far-Red (%61: %20: %15: %4)		16 h	150, 200, 250 μ mol m−2 s−1 PPFD	Peat substrate		21–17 °C	%65 relative humidity	
Green basil (cv.Sweet Large) and Purple basil (cv.Dark Opal)
(Skowron et al., 2025)	LED	661 nm, 633 nm, 520 nm, 434 nm (no information about the colors)	16 h	200 μ mol m−2 s−1 PPFD	Hydroponics		23 °C during the day, 20 °C at night	%50–55 relative humidity	
Lentil (Lens culinaris)
(Silva et al., 2024)	LED	Cool white: 400–700 nm
Red: 600–700 nm
Blue: 400–500 nm
RBW: 70.5% red, 8.5% green, 21.0% blue	12 h	250 μ mol m−2 s−1 PPFD for the constant regime / 85-500 m−2 s−1 PPFD for the Gaussian regime	Hydroponics		24 °C	%60 relative humidity	
Notes.

LED light-emitting diode

PPFD photosynthetic photon flux density

Some procedures that can be applied to microgreens before and after harvest can also increase their nutritional and functional values (Frąszczak & Kleiber, 2022). Sun et al. (2015) reported that preharvest application of calcium chloride (CaCl2) to broccoli microgreens was associated with a higher glucosinolate level. In another study, Kim et al. (2024) treated broccoli microgreens with various concentrations of illite, a natural mineral powder, during the germination phase. They reported significant increases in Ca, P, and K mineral contents, as well as in vitamin C and free amino acids. In addition, the total weight of the microgreens increased markedly compared with those not treated with illite, and the sulforaphane content rose by up to 47%. In addition, microgreens have a very short shelf life (Mir, Shah & Mir, 2017). Paradiso et al. (2018) examined the shelf life of microgreens packaged using the modified atmosphere packaging method and reported that freshly cut microgreens had an average shelf life of 10 days at 5 °C. With the developing technology, some ergonomic and environmentally friendly new packaging techniques can preserve the nutritional quality of microgreens and extend their shelf life (Ghoora & Srividya, 2020). One study demonstrated that hot air drying (up to 95 °C) was more suitable than freeze-drying for extending the postharvest shelf life of microgreens, broadening their potential applications, and preserving their nutritional value. It was also reported that microgreen powder could be incorporated into formulated foods (Jauregui et al., 2025). In another study, Gudziṅskaite et al. (2024) found that postharvest storage of mustard and kale microgreens under light conditions (250 PPFD) was more advantageous for maintaining nutritional value and shelf life compared with dark storage. The study further indicated that postharvest microgreens frozen with liquid nitrogen retained the highest β-carotene content (at day 150 for kale and day 200 for mustard) under high light intensity compared with storage in the dark. Therefore, storage conditions were shown to have a substantial influence on shelf life and nutrient preservation.

Potential and current food safety risks in microgreens

Many environmental factors such as soil, irrigation water, seed storage conditions, plant nutrient solution, worker health/hygiene, and facilities and buildings in microgreens that are generally consumed raw can cause contamination with highly dangerous pathogens through cross-contaminations (Bergšpica et al., 2020; Isik, Cetin & Topalcengiz, 2024; Wright & Holden, 2018). Especially in plants grown hydroponically, it is usual for viruses present in the feed water to pass into plant tissues (Fuzawa et al., 2021). Bacterial pathogens such as Salmonella, Escherichia coli, and Listeria monocytogenes are considered the primary bacterial agents for contamination of microgreens (Isik, Cetin & Topalcengiz, 2024). Therefore, post-harvest washing, or sanitation applications may be inadequate microbiologically (Reed et al., 2018; Fuzawa et al., 2021).

Reed et al. (2018) investigated the extent to which alfalfa sprouts and Swiss chard microgreens grown from seeds contaminated with two Salmonella enterica serotypes (Hartford and Cubana) are affected by various external factors such as seed storage time and growth environment. They reported that irrigation water and seed storage time had significant effects on the development of Salmonella enterica serotypes in sprouts and microgreens.

Naushad et al. (2022) reported that Listeria monocytogenes strains were isolated from some microgreen samples grown in Canada. Xiao et al. (2015) studied E. coli populations in edible and inedible parts and substrates by hydroponically growing white radish seeds (Raphanus sativus var. Longipinnatus) contaminated with E. coli in peat moss (also known as peat) and polyethylene pads. It was observed that E. coli significantly survived and multiplied in various tissues of the plant in both types of cultivation, with higher levels in the hydroponic medium. Xiao et al. (2014) investigated the contamination status of sprouts and microgreens grown from white radish seeds contaminated with E. coli. Bacterial growth was reported in both sprouts and microgreens, but significantly less in microgreens, in the study. Wang & Kniel (2015) examined the ability of the virus to attach to the edible tissues of microgreens by contaminating hydroponically grown cabbage and mustard seeds with murine norovirus. In the study, they reported that the virus could hold on to both roots and edible plant tissues and maintain its stability for 12 h after harvest, gradually decreasing on the 8th and 12th harvest days. In another study, daikon, red cabbage, broccoli, and mustard microgreens grown in water contaminated with low and high concentrations of Salmonella enterica, E. coli, and Listeria monocytogenes were examined for food safety. The findings indicated that enteric pathogens can migrate from contaminated water sources into the edible inner tissues of microgreens, highlighting the importance of maintaining safe growing conditions to prevent contamination (Rao, Pradhan & Patel, 2025).

In light of these studies, it is evident that any microbial contamination caused by environmental factors during the entire production process in microgreens can cause serious health problems. This situation creates the need for appropriate sanitation, especially for hydroponically grown microgreens. Therefore, it is important to take all food safety precautions at the beginning of production. Cleanliness of the seeds, substrate, plant nutrient solution and irrigation water to be used will greatly reduce food safety risks.

Conclusions

Global changes in nutritional behaviors and trends, food insecurity, malnutrition problems, and above all, climate change and the disadvantages of modern agriculture such as agricultural pesticide use, labor and financial losses increase the need for healthy alternative foods. Microgreens offer a convenient option for individuals without the need for professional equipment, extensive fields, or chemical fertilization. The popularity of microgreens is on the rise, attributed to their high nutritional profile and rapid growth cycle. Very few animal experiments and cell culture studies point to various health benefits of microgreens, which are rich in polyphenols and antioxidants, and present them as nutraceuticals. Also, no toxic effects related to microgreens have been reported in the current literature. On the other hands, studies have shown that the phytochemical and antioxidant properties of microgreens are affected by factors, such as light conditions, substrate, temperature, and seed genotype. The dimensions of these effects remain unclear and more studies are needed to obtain optimum efficiency from microgreens production. On the other hand, microbial contamination caused by environmental factors in microgreens is a critical issue, and therefore appropriate sanitation methods should be developed. It is recommended that future studies focus on these critical points.

Supplemental Information

Supplemental Information 1 References dataset

Additional Information and Declarations

Competing Interests

Author Contributions

Data Availability

The authors declare there are no competing interests.

Seyda Kaya conceived and designed the experiments, performed the experiments, analyzed the data, authored or reviewed drafts of the article, and approved the final draft.

Hülya Yardımcı conceived and designed the experiments, performed the experiments, analyzed the data, authored or reviewed drafts of the article, and approved the final draft.

The following information was supplied regarding data availability:

This is a literature review, hence it did not utilize raw data.

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
