# Peer review of "Microgreens: nutritional properties, health benefits, production techniques, and food safety risks"

_PeerJ, doi:10.7717/peerj.17938_

## Round 0.1 · original submission · Major Revisions

· Academic Editor

Major Revisions

Please authors, two reviewers have attended to your work, and observed a number of areas that need your attention.

Please provide ample detail, especially in responding to comments, not only in the revised manuscript.

Please also provide a strong ‘methodology’ section, that elaborates on the various steps that were employed to perform the literature synthesis. Remember to elaborate on your inclusion and exclusion criteria, provide how you extracted various information, steps you took to construct tables.

Please endeavor to provide more schematic diagrams, to support the key areas of the body, as well as the concluding section.

Look forward to your revised manuscript.

**Language Note:** The review process has identified that the English language must be improved. PeerJ can provide language editing services - please contact us at [email protected] for pricing (be sure to provide your manuscript number and title). Alternatively, you should make your own arrangements to improve the language quality and provide details in your response letter. – PeerJ Staff

·

Basic reporting

The topic is a relatively novel and emerging concept of great importance and the basic reporting was ok to a very large extent but a few sentences were incomprehensible and were noted in the manuscript. Language editing will improve the manuscript.

The rationale and objective of the review needs to be clearly captured in the abstract

If available in published articles, more details on the uses of Microgreens should be added in lines 95, 96 and 97 in the manuscript.

For the section on 'Nutritional Properties of Microgreens': This section has good details regarding micronutrients content of microgreens but did not capture specific amounts/concentrations of the micronutrients. So, the addition of the specific amounts/concentrations of the micronutrients in line 155 of the manuscript will improve the quality of the article.

Experimental design

The review has enough and well-articulated details relevant to the title and the Survey Methodology. But the section on NUTRITIONAL PROPERTIES OF MICROGREENS, lacks details on proximate composition as an aspect of nutritional composition. Please include details on proximate composition if there are pieces of information on the proximate composition of microgreens from various crops in various Journal papers consulted as the article was compiled?

A few citations of references are needed in the manuscript and were pointed out in the annotated manuscript.

The first paragraph (line 117 – 124) under the section on ‘Nutritional Properties of Microgreens’ should be transferred to the ‘Production Techniques’ section.

The sub-section on 'Current food safety risks in Microgreens' could be re-captioned to read - 'Potential and current food safety risks in Microgreens' to accommodate the fact that there exists the potential of possible food safety risks from microgreens from some crops which have not yet be researched on.

The section on 'Production techniques of microgreens' properly captured details on the lightening, photo-period and wavelength for microgreens production but had limited detail regarding cultivation on soil as against biopolymer-growing fibers or hydroponically produced microgreens as well as their merits and demerits.

Validity of the findings

The inferences and conclusions are supported by the text in the main body of the Review paper.

Additional comments

Please, the entire text in the document should be ‘justified’; the entire text in the document should be evenly distributed between the margins.

·

Basic reporting

This review paper attempted to provide an overview of the nutritional properties, health benefits, production techniques, and food safety risks associated with microgreens which can be produced from a variety of plants. My comments are as follows.

Keywords
Increase the number of keywords to at least 5.

Abstract
L 27 – 35 should be part of the introduction in a way.
The abstract should be rewritten completely as it does not provide a true reflection of the review paper. It should clearly reveal the background of the study, scope, objective, study approach, key findings, and conclusions.

Introduction
This section lacks clarity in defining the following terms; sprouts, baby green, microgreen, and mature green. The authors just jumped into talking about them making the literature incoherent. What are the key differences between them?
The statement of the problem was not clearly stated nor was an explanation given for selecting microgreens as the subject matter from the progressive growth stages of plants. More literature synthesis is needed to expatiate the rationale behind this review.
In Line 77, you mentioned that microgreens are grown hydroponically, does it mean that there are no microgreens that are grown in the soil? Please clarify and incorporate it in the text.
There is no figure or picture of microgreens from plants.

Experimental design

Methodology
This section is not so clear. What are your search strategies? Like inclusion and exclusion criteria, years of study considered, etc. The inclusion and exclusion criteria should be decided based on the objective or research question of the review, which would inform the search strategy rather than the other way around.
L 124: Kindly state what was analysed and in what way were they analysed. Methods and outcomes if possible. State the data you will be extracting and how the data will be synthesised.

Validity of the findings

Nutritional properties of microgreens
This section was well written but to enhance the quality of this section, kindly prepare a nutritional profile table showing a comparison between microgreens, seeds, baby green and/or mature green.
L 137 – 138: What is super nutrient? You mentioned that microgreens are also called super nutrients because they are rich in antioxidants, minerals, vitamins, etc. There is no data or table to show how true this claim is in comparison with other normal plants or vegetables that also have these nutrients. This will provide published evidence to buttress your argument on microgreens.

Food safety risk
Kindly prepare a table showing the potential risks of different microgreens and suggested solutions with references.
Highlight how to ensure the safety of homegrown microgreens.

Conclusion
L 469 – 470: What do you mean by disadvantages of modern agriculture? This was not mentioned anywhere in the manuscript.
Overall, the conclusion should be rewritten. This section did not succinctly summarize the key points and findings discussed in the review paper which entails a brief recap of the main arguments/research question/objective and evidence presented throughout the paper. It also failed to state the significant contributions and implications of the reviewed research. The author did not clearly offer suggestions for future research directions or areas that need further exploration based on gaps identified in the reviewed literature.

---

## Round 0.2 · Minor Revisions

· Academic Editor

Minor Revisions

Please, authors, kindly attend to the comments raised.
Authors, kindly implement the following in addition:

1) Sections "Nutritional properties of microgreens" to " Potential and current food safety risks in microgreens" should be italicized, and all be embedded into "Discussion of synthesized literature"... This makes the entire work have four major sections only, namely: Introduction, Survey methodology, Discussion of synthesized literature, Conclusions.

2) Please, kindly brainstorm and come up with a figure that summarizes "Production techniques of microgreens"...and refer to this figure in the text when you mention the techniques.

3) Please, the section "Potential and current food safety risks in microgreens" is somewhat scanty. Kindly use this revision opportunity to add more information, tell us more about "why?", not just "what?"

4) Please, provide more discussion of Table 1, very important.

Look forward to your revised manuscript. Thank you

·

Basic reporting

Yes, the rationale and objective of the review has now been captured in the abstract.

However, in line 30, the core nutritional advantage of microgreens; being good sources of minerals, vitamins, antioxidants and phenolic compounds could be specifically captured in the abstract.

Please delete “And” in line 31.

Also in line 40, the statement “will shed light on future studies” appears inappropriate. Authors may reconsider that statement and rephrase it as to better represent what they intended to express.

In line 103, “was investigated” could be replaced with were documented.

Please add a citation in line 160 after the statement “In a study by ………”

Experimental design

The review has enough well-articulated details relevant to the title. But the section on NUTRITIONAL PROPERTIES OF MICROGREENS, lacks details on proximate composition as an aspect of nutritional composition. Is there not even a piece of information in various Journal papers on the proximate composition of microgreens from various crops?

This was noted in the initial review but it was not handled.

Nutritional properties of microgreens will involve major aspects of nutritional composition which include minerals, vitamins and proximate composition (crude fibre, crude protein, ash, carbohydrate, lipid and moisture).

Even though the authors captured phytochemicals, minerals and vitamins in microgreens, it may seem inappropriate to use the subtitle – Nutritional properties of microgreens, as it is a broad term and didn’t capture proximate composition parameters. Its very likely that the crude fibre and ash content for microgreens will differ from that of mature plants.

An article with DOI: https://doi.org/10.1016/j.fufo.2023.100262 (K. S. Arya, M. Sangeeta Kutty, T. Pradeepkumar. Microgreens of tropical edible-seed species, an economical source of phytonutrients- insights into nutrient content, growth environment and shelf life,
Future Foods, Volume 8, 2023) touched on crude protein and crude fibre of microgreens. Authors could search for such articles and add such details to make the article superb.

In line 125, please add “sources” after “nutrients”.

Yes. Citations have been included to sentences/sections that lacked citations.

The Title of the figure should be written below the figure and appropriate citation of its source(s) should also be provided.

The units in Table 1 should be put in brackets and written only in the table header.

The text “Table 1:” should be on the same line with the title of the table.

Validity of the findings

No comment

·

Basic reporting

The manuscript has been improved significantly. Thanks to the authors but a few concerns remain.

L60: incoherent statement

L31; L86; L216: You can start a sentence with `And`

Experimental design

No comment

Validity of the findings

Table 1: Why are some texts and values highlighted? Write carb in full, i.e., carbohydrate. Prepare a scientific Table by removing vertical lines from the Table and putting 3 horizontal lines (2 on top and 1 at the bottom) appropriately.

Botanical names and names of microorganisms should be italicized.

L437-439: See annotated manuscript.

---

## Round 0.3 · Minor Revisions

· Academic Editor

Minor Revisions

Per earlier discussion, please take this opportunity to add the additional references that are relevant to properly update this article.

Please make sure that you upload both a clean and tracked changes manuscript that are identical in content, only showing the changes clearly in the tracked changes manuscript, when you resubmit.

The Academic Editor will need to review the revision to confirm they agree with the additions and do not feel that other references are necessary.

---

## Round 0.4 · accepted · Accept

· Academic Editor

Accept

Very satisfied with the revised manuscript. It is now acceptable for publication. Thank you, authors, for your patience throughout the peer review process, and for diligently addressing reviewers' (and editors') concerns. We appreciate your finding PeerJ as your journal of choice and look forward to your future scholarly contributions. Congratulations :)